# Continual Learning for Instruction Following from Realtime Feedback

**Alane Suhr**
University of California, Berkeley*
suhr@berkeley.edu

**Yoav Artzi**
Cornell University
yoav@cs.cornell.edu

## Abstract

We propose and deploy an approach to continually train an instruction-following agent from feedback provided by users during collaborative interactions. During interaction, human users instruct an agent using natural language, and provide realtime binary feedback as they observe the agent following their instructions. We design a contextual bandit learning approach, converting user feedback to immediate reward. We evaluate through thousands of human-agent interactions, demonstrating 15.4% absolute improvement in instruction execution accuracy over time. We also show our approach is robust to several design variations, and that the feedback signal is roughly equivalent to the learning signal of supervised demonstration data.

## 1 Introduction

The dynamics that arise in situated language interactions between human users and automated agents expose a plethora of language learning signals. A prime example of such a signal is explicit feedback: when users convey their intent via natural language instructions for an agent to follow, they are well positioned to provide feedback, for example, through a stream of binary signals as the agent acts.

Learning from this type of signal has significant potential. It shifts the burden of learning from annotated data to learning through interaction with users, which not only reduces data costs, but also enables continual[2] improvement through interaction with users. This signal also fundamentally differs from gold-standard annotated data: it directly targets the current agent behavior, rather than reflecting optimal human behavior that may be of less relevance to the agent's current policy. This approach stands in contrast with most methods for learning to follow instructions, where the use of demonstration data entails high data costs, and the clear separation between training and deployment passes over opportunities for learning through interaction with users [e.g., 3, 1, 38].

In this paper, we study the problem of continually[2] improving an automated instruction-following agent[3] by learning from user feedback in human-agent interactions. We situate the interaction in a collaborative environment, which provides conditions necessary for this type of learning: users are incentivized to improve the agent's capabilities by providing feedback, and are continuously present as the agent executes their instructions. We use the CEREALBAR scenario [41], where two participants collaborate towards a common goal in a shared world, using natural language instructions to coordinate their actions. During interaction, human users write instructions for an agent, and provide feedback to the agent as it executes them via a stream of binary signals. Figure 1 illustrates

---

*Work done while at Cornell University.

[2]We use the term *continual learning* to refer to a learning setting where an agent continually improves its task performance [14], in our case following instructions. The term is used at times to describe the domain adaptation challenge of continually learning to handle new tasks. Our work does not address this problem.

[3]Throughout the paper, we use *agent* to refer to an automated instruction-following system.

37th Conference on Neural Information Processing Systems (NeurIPS 2023).

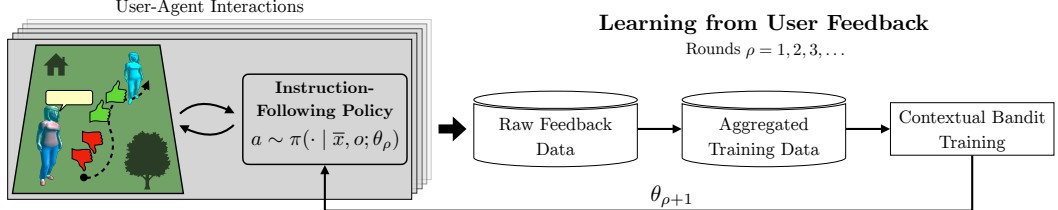

Figure 1: Illustration of our continual learning process.[2] The process progresses in rounds $\rho$, each including deployment and training. In deployment, we sample and execute actions $a$ from a policy $\pi$ parameterized by $\theta_\rho$, conditioned on user-written instruction $\overline{x}$ and agent observation $o$. Concurrently, users provide binary feedback as they observe instruction execution. We convert this feedback to rewards, and estimate parameters $\theta_{\rho+1}$ of next round's policy with a contextual bandit objective.

the setup and our learning process. To the best of our knowledge, our work is the first to demonstrate continual learning from human feedback for natural language instruction following.

A key challenge is the complexity of the learning signal. Users provide feedback at their discretion and under time pressure as the agent executes their instructions, resulting in an unpredictable, and often noisy, learning signal. While the concurrency of agent actions and user feedback provides a weak alignment, delays in human response and lack of clarity about which action or actions they critique make it challenging to attribute a learning signal to specific actions. Further complicating learning is the feedback loop created by the continually changing agent behavior and constant adaptation by human users. These factors create a scenario that is particularly difficult for complex methods as they often make various assumptions about the learning signal and the interaction dynamics [20, 31]. We opt for a simple approach with minimal assumptions, and formulate learning as a contextual bandit scenario. We alternate between deployment, where the agent interacts with users that provide feedback, and training, where we compute immediate rewards from feedback signals, and weigh examples using the current policy to account for the continually changing agent behavior.

We experiment with our approach through interactions with humans users, where we repeatedly deploy, train, and re-deploy the agent. We study the effects of human adaptation and process design decisions on the efficacy of learning. We design our experiments to tease apart genuine agent improvements from human user adaptation, and observe dramatic improvements in agent behavior beyond what user adaptation explains, 15.4% absolute improvement in instruction execution accuracy in our longer running experiment. Our code and data is released under the MIT license 2023.[4]

## 2   Technical Overview

**Interaction Scenario** We situate user-agent interactions in CEREALBAR [41], a collaborative scenario where two participants, a leader and a follower, collect sets of matching cards together in a shared, 3D environment. The two participants collaborate towards a shared goal, act in the world, and coordinate using natural language. The leader plans what the pair should do, acts according to the plan, and describes the follower's part of the plan using natural language instructions. The follower's role is to follow the instructions. The pair receives a point for each set of cards they successfully collect. Their goal is to maximize their score. We add to the original CEREALBAR setting the ability for the leader to provide binary feedback signals to the follower as they execute instructions. In our study, the agent controls the follower and the leader is always a human user. We use CEREALBAR as it is the only research environment designed to support the real-time collaborative interaction scenario we study: alternatives (e.g., Thor [23]) are not designed to deploy at scale for model-user interaction studies, and would require extensive modifications to support multi-agent collaborative interactions.

**Task** The agent's task is to map natural language instructions and observations to follower actions. Actions include moving FORWARD or BACKWARD, turning LEFT or RIGHT, and completing instruction execution with STOP. The agent observation is a partial view of the world state from the

---

first-person perspective of the follower. Formally,[5] given an instruction $\overline{x}$ and an initial observation $o_1$, our goal is to generate a sequence $\langle (o_1, a_1), \ldots, (o_m, a_m) \rangle$, where $o_i$ an agent observation, $a_i$ is the action taken, and $a_m = \texttt{STOP}$.

[4]Our code and data is available here: `https://github.com/lil-lab/clif_cb`. The original CEREALBAR game and data [41] is licensed under MIT License 2022.

**Inference and Learning** We define the follower agent as a policy $\pi(\overline{x}, o; \theta)$ parameterized by $\theta$ that maps instruction $\overline{x}$ and state observations $o$ to a distribution over actions.[6] During interaction with users, for each user-written instruction $\overline{x}$, we sample and execute actions $a_i \sim \pi(\cdot \mid \overline{x}, o_i; \theta)$ until we sample the instruction completion action. Concurrently, the user may provide positive and negative feedback signals $f$. Feedback signals are given in realtime, and are not timed to coincide with a specific action. Between any two agent steps, there could be any number of feedback signals.

We optimize parameters through rounds of continual learning (Figure 1). At each round $\rho$, we deploy the policy with parameters $\theta_\rho$ to interact with users, and then estimate parameters $\theta_{\rho+1}$. During deployment, we collect agent instruction executions and user feedback. The execution of an instruction $\overline{x}$ during interaction creates a trace made of two sequences $(\langle (o_i, a_i, w_i^a) \rangle_{i=1}^m, \langle (f_j, w_j^f) \rangle_{j=1}^n)$, where each action $a_i$ and feedback signal $f_j$ are paired with wall times $w_i^a$ and $w_j^f$. We compute reward from the feedback to construct training examples $(\overline{x}, o, a, r)$, where $r$ is a reward. We optimize policy parameters by solving a contextual bandit problem, maximizing immediate expected reward.

**Evaluation** Our main metric is instruction execution accuracy for user-written instructions. Because we only have access to agent instruction executions, and not ground-truth demonstrations, we cannot compute accuracy automatically. We use crowdsourcing to acquire human judgments of follower accuracy after each round of deployment. We also evaluate trends in user feedback and ratings.

# 3 Continual Learning

We estimate the policy parameters from user feedback during interaction with users. The process, illustrated in Algorithm 1, progresses in rounds. Each round $\rho$ includes: (a) deploying the agent policy parameterized by $\theta_\rho$ to interact with users (Section 3.1), (b) computing rewards from user feedback to construct training data $\mathcal{D}_\rho$ (Section 3.2), and (c) optimizing the policy parameters using all data observed so far $\bigcup_{\rho'=0}^{\rho} \mathcal{D}_{\rho'}$ to compute $\theta_{\rho+1}$ (Section 3.3). We initialize the process with a policy parameterized by $\theta_1$ estimated on human demonstration data $\mathcal{D}_0$ (Algorithm 1, Line 1).

## 3.1 Deployment Interactions

During deployment in round $\rho$, users collaborate with the agent, and delegate to it tasks by typing natural language instructions. For each instruction $\overline{x}$, we sample a sequence of actions from the policy $a_i \sim \pi(\cdot \mid \overline{x}, o_i; \theta_\rho)$ (Lines 9–15). Executing an action in the environment results in a new observation $o_{i+1}$ (Line 11). For each action, we record the wall time $w_i^a$ as the time when the user starts seeing the follower starting to execute the action (i.e., the action animation starts) (Line 12). We terminate instruction execution when the stop action $\texttt{STOP}$ is sampled (Line 15).

At any point in time during the agent's instruction execution, the user may provide binary (positive or negative) feedback signals, creating a stream $\langle (f_j, w_j^f) \rangle_{j=1}^n$ where $w_j^f$ is the user's wall time when a feedback button is pressed (Line 13). We also allow the user to reboot the follower at any point during an instruction execution (omitted from Algorithm 1). This is meant to terminate very bad executions before they damage the interaction, for example by leading the follower too far astray. Rebooting adds a negative feedback signal to the end of the stream, and removes all queued instructions.[7]

The execution of an instruction by the agent and the feedback given by the user are combined together to create a trace made of two sequences $\langle (o_i, a_i, w_i^a) \rangle_{i=1}^m$ and $\langle (f_j, w_j^f) \rangle_{j=1}^n$.

---

[5]Suhr et al. [41] provide complete formal notation for CEREALBAR. We use a simplified notation to focus on the follower instruction following task only.

[6]Our policy takes as input a single instruction. Considering interaction history, including previous instructions, is an important direction for future work.

[7]CEREALBAR allows the leader to queue multiple instructions; the follower only sees the current instruction.

**Algorithm 1** Continual learning for instruction following from realtime user feedback.

**Input** Human demonstration data $\mathcal{D}_0$; number of interactions per round $T$; dataset construction process `convert_feedback_to_reward` that maps from an instruction and feedback-annotated execution trace to a set of reward-annotated training examples (Section 3.2); learner `train_policy` that optimizes a contextual bandit policy gradient objective (Section 3.3).

1: $\theta_1 \leftarrow$ `train_policy`$(\mathcal{D}_0)$         ▷*Initialize policy parameters with human demonstration data.*
2: **for** round $\rho = 1, \dots$ **do**
3:      $\mathcal{D}_\rho \leftarrow \{\ \}$
4:      **for** $T$ interactions **do**                  ▷*Deploy the current policy for $T$ interactions.*
5:          **repeat**
6:             $\overline{x}, o_1 \leftarrow$ observe instruction from user and current world state
7:             $\overline{a}, \overline{F} \leftarrow \langle\ \rangle$            ▷*Initialize empty action and user feedback sequences.*
8:             $i \leftarrow 1$
9:             **repeat**                 ▷*Generate actions for instruction.*
10:                $a_i \sim \pi\left(\cdot \mid \overline{x}, o_i; \theta_\rho\right)$         ▷*Sample action from policy.*
11:                $\begin{cases} o_{i+1} \sim \mathcal{T}(o_i, a_i) \\ \overline{f}_i \leftarrow \text{observe feedback} \end{cases}$     ▷*Concurrently: execute action and observe feedback.*
12:                $\overline{a} \leftarrow \overline{a} + \langle(o_i, a_i, \texttt{walltime()})\rangle$
13:                $\overline{F} \leftarrow \overline{F} + \overline{f}_i$         ▷*Collect user feedback given during action execution.*
14:                $i \leftarrow i + 1$
15:             **until** $a_{i-1} = $ `STOP`
16:             $\mathcal{D}_\rho \leftarrow \mathcal{D}_\rho \cup$ `convert_feedback_to_reward`$\left(\overline{x}, \overline{a}, \overline{F}\right)$
17:          **until** interaction terminates
18:      $\theta_{\rho+1} \leftarrow$ `train_policy`$\left(\bigcup_{\rho'=0}^{\rho} \mathcal{D}_{\rho'}\right)$         ▷*Train policy for next deployment using all data.*

## 3.2 Dataset Construction

We use all traces collected in round $\rho$ to construct a training dataset $\mathcal{D}_\rho$. Each example in $\mathcal{D}_\rho$ is a tuple $(\overline{x}, o, a, r)$, where $\overline{x}$ is an instruction written by the user in round $\rho$, $o$ is the agent observation when action $a$ was selected during the execution $\overline{x}$ in the interaction, and $r$ is a numerical reward we compute from the feedback stream. For each instruction written during round $\rho$ we may create no examples if no feedback was given, or multiple examples if computing the rewards from feedback resulted in attributing rewards to multiple actions.

**Simple Reward** We consider the value of each positive feedback as $f = +1$ and each negative feedback as $f = -1$. We compute the reward $r_i$ for an action $a_i$ as the sign of sum of feedback signals that occur between it and the next action after correcting for human response delay time [31]:

$$r_i = \operatorname{sgn} \sum \left\{ f_j \mid w_i^a < w_j^f - d \leq w_{i+1}^a \right\} \ , \tag{1}$$

where $w_i^a$ is the wall time of action $a_i$, $w_j^f$ is the wall time of feedback signal $f_j$, and $d$ is a human response delay constant ($d = 0.2$ seconds). If the set is empty with no feedback signals within the time window, or if $r_i = 0$, we do not set a reward or create an example for the corresponding action.

**Heuristically Propagated Reward** As users are not required to follow any specific feedback schedule, the simple reward computation is likely to result in relatively sparse reward assignment, with many actions not receiving any learning signal and thus essentially thrown out as training data. Indeed, in our experiments, we observe that roughly only 63.1% of actions receive feedback. However, human feedback may not be about the most recent action only, but can potentially be attributed to multiple recent actions. For example, the user may not be certain if to discourage a behavior, such as turning and going in a certain direction, until the trajectory is clear after several steps.

We address this by taking inspiration from the use of eligibility traces for credit assignment [42], and create heuristics to propagate the reward to actions that otherwise receive no reward. If an action is assigned no reward, we propagate to it a reward from a later action by assigning to it the reward of the next action that is assigned one, if one exists within the next eight actions. The only exception is that we do not propagate reward from a `STOP` action that receives a negative reward.

We additionally prevent noisy feedback from being propagated by considering the scenario dynamics. We remove all training examples for actions following an action that both results in a card selection (or de-selection) and receives a negative reward. If an action with positive reward results in an invalid set (i.e., the selected cards do not match), we remove it from the training data and do not propagate its award. These scenario-specific heuristics were developed by identifying common feedback errors in the data. Following propagation, we observe that 82.2% of the actions have reward assigned to them, up from 63.1, without increasing the reward error rate of about 7.0%.[8]

### 3.3  Parameter Optimization

The complexity of human feedback complicates reward attribution. We adopt a simple approach and maximize the immediate reward (i.e., a discount factor of zero).[9] This creates a contextual bandit scenario. Because of the costs of human interaction and our aim to rapidly improve through few interactions, we emphasize simplicity and use a REINFORCE-style policy gradient objective [47]. At the end of each round, we train from scratch using all the data observed so far $\bigcup_{\rho'=0}^{\rho} \mathcal{D}_{\rho'}$. We process the initial human demonstration data $\mathcal{D}_0$ to the same form as the reward data we get from user interactions, so we can use it seamlessly with the data we get from the rounds. For each action $a$ and input instruction $\overline{x}$ and observation $o$ in $\mathcal{D}_0$, we create an example $(\overline{x}, o, a, +1)$; we consider all demonstration data as gold-standard, and set the action reward to $+1$.

During round $\rho$, the gradient for an example $(\overline{x}, o, a, r)$ from dataset $\mathcal{D}'_\rho$, $\rho' \leq \rho$ is:

$$\Delta = c(a, \overline{x}, o, \rho') r \nabla \log \pi\left(a \mid \overline{x}, o; \theta\right) \tag{2}$$

$$c(a, \overline{x}, o, \rho') = \begin{cases} \min\left(1, \dfrac{\pi\left(a \mid \overline{x}, o; \theta\right)}{\pi\left(a \mid \overline{x}, o; \theta_{\rho'}\right)}\right) & \rho' > 0 \\ 1 & \rho' = 0 \end{cases},$$

where $c(\cdot)$ computes a clipped inverse propensity scoring (IPS) coefficient of the policy gradient [17], and $\theta_{\rho'}$ are parameters with which the examples in $\mathcal{D}_{\rho'}$ were generated. IPS is commonly used for debiasing policies during off-policy learning. We also use IPS to avoid the problem of exploding gradients while training on negative examples ($r < 0$) [22]. We clip the value of the IPS coefficient to a maximum of 1 to avoid overfitting on positive examples [12].

## 4  Experimental Setup

**Initialization Data** Except when specified otherwise, the demonstration training dataset $\mathcal{D}_0$ includes 8,790 instructions from 456 randomly-sampled human-human interactions from Suhr et al. [41]. Pilots studies showed estimating $\theta_1$ using this data provides a good balance of human-human data amount and initial system performance for interaction with human users.

**Model** We implement the policy $\pi$ as a neural network with parameters $\theta$. The network design is based on the design of Suhr et al. [41], except that we account for partial observability. Roughly, the instruction $\overline{x}$ and observation $o$ are embedded independently. We use instruction-conditioned convolutions on the embedded observation to mix features from both inputs, and a modified LINGUNET [6] generates a distribution over actions. Appendix B provides formal model details.

**Deployment** In each round of deployment, we collect up to a fixed number of interactions per user. We instruct users to use the reboot button sparingly, and only when they anticipate the follower will make a mistake that cannot be easily corrected. We do not enforce any specific feedback patterns. Users are aware that their instructions and feedback will be used to train the agent for future rounds.

**Evaluation** The main measure of performance is the agent's instruction execution accuracy during interactions with human users. Because we have no human demonstration or annotation for this data, we perform post-hoc manual evaluation for each round. We randomly sample instructions from each round. We only sample instructions the agent completed (i.e., executed the STOP action) to annotate.

---

[8]Error rate determined by manually annotating actions and feedback from 100 randomly-sampled instructions.
[9]We consider reward propagation (Section 3.2) to be part of the mapping of feedback to reward, so not considered as discounting in the learning objective.

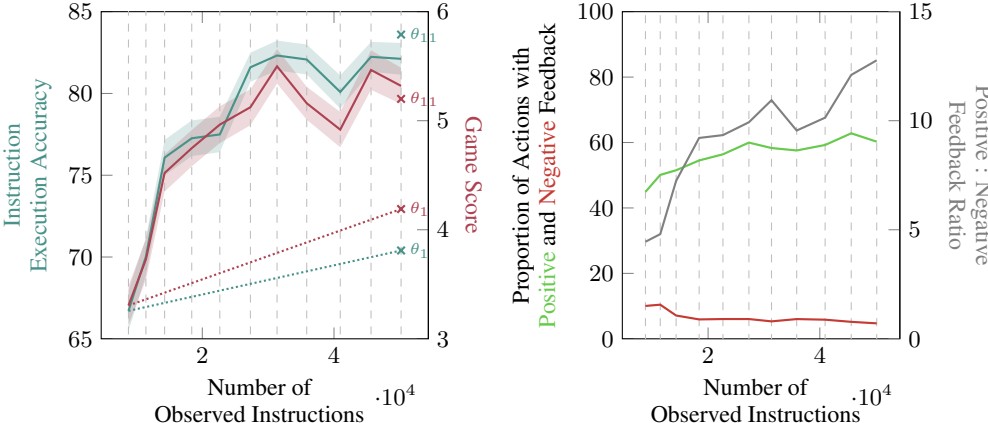

Figure 2: *Left*: Mean estimated instruction execution accuracy and game scores across 11 rounds. The $x$-axis shows the number of instructions observed. We mark with × the accuracies and scores from the post-hoc study of user adaptation for $\theta_1$ and $\theta_{11}$. Dotted lines illustrate the share of the improvement due to user adaptation. *Right*: Proportion of actions receiving positive and negative feedback from users over 11 rounds, and the ratio of frequency between positive and negative feedback. The x-axis shows the number of instructions observed. In both plots, dashed lines show round boundaries.

We assume all instructions rebooted by the user are failures, and adjust down the accuracy as estimated from the annotations according to the ratio of rebooted instructions. Appendix D formally defines the evaluation metric. We show the instruction, an animation of the agent's execution, and a list of cards that were reached by the follower during execution. We ask whether the follower correctly executed the instruction, and whether the follower made any number of errors in a pre-defined list. For performance measurements and statistics, we report the mean and confidence interval,[10] except where noted. When emphasizing the relationship between the amount of training data and each measure, we plot the metric by the number of observed instructions rather than by round index.

**Crowdsourcing** We use MTurk for both user-agent interactions and post-hoc instruction evaluation. We recruit workers from English-majority locales using a qualification quiz and tutorial. Appendix E contains crowdsourcing management and compensation details. We maintain novice and expert groups of workers. Expert workers have higher compensation, and access to the post-hoc instruction accuracy tasks. To become an expert, workers must show they understand the task and annotation expectations during interactions in the novice pool. We qualify 108 workers, including 65 expert workers. During deployment, we balance the number of interactions between workers by limiting each worker to about eight interactions per agent and round.

## 5 Results and Analysis

We conduct two experiments: an 11-round experiment to observe long-term learning and user behavior trends, and a shorter 5-round experiment to compare several learning design decisions.

### 5.1 Long-Term Experiment

We evaluate over eleven rounds of deployment and training. This experiment uses the heuristically propagated reward, with the goal of obtaining more training examples compared to the simple reward (Section 3.2). In total, we collect 3,368 games and 46,573 instructions, at a cost of $15,944.45 USD.

**Agent Accuracy** Figure 2 (left) shows instruction execution accuracy (estimated with 800 instructions per round) and number of points across the eleven rounds. Accuracy improves from $66.7 \pm 1.5$ in the initial round to $82.1 \pm 1.1$ in the final round. The continual learning process also results in increased

---

[10]Confidence intervals are computed using bootstrap sampling. $n = 10{,}000$ samples for interaction-level metrics, and $n = 1{,}000$ samples for instruction- and action-level metrics.

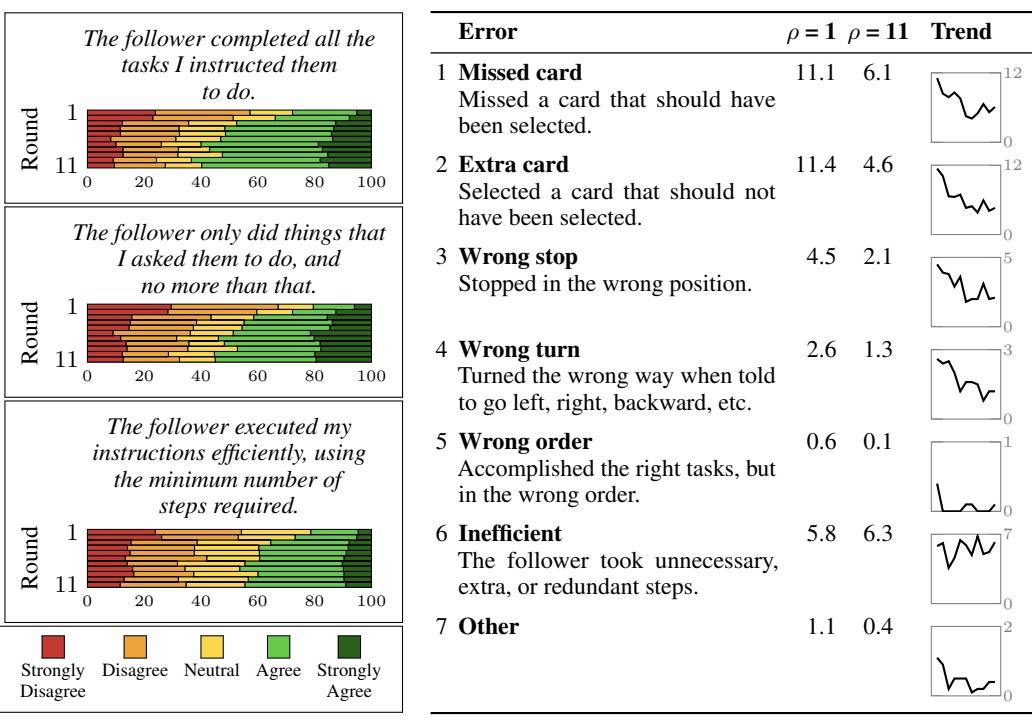

Figure 3: *Left*: Distribution over interactions of post-hoc user agreement with three statements about agent performance. *Right*: Instruction execution errors during 11 rounds for seven common categories. Center columns show prevalence of each error type as a proportion of analyzed instructions, for the initial ($\rho = 1$) and final ($\rho = 11$) agents; the rightmost column shows trend in prevalence over time.

interaction success: the game score increases from $3.3 \pm 0.2$ to $5.3 \pm 0.2$ over time. Performance begins to plateau after the sixth round, which we hypothesize is due to model performance limitations. Appendix F provides additional evaluation on a static, held-out set of human demonstration data.

**Influence of User Adaptation on Performance** User adaptation over time is a confounding factor, as improvements in agent performance may be due to users adapting what tasks they delegate to the follower agent, or how they phrase their instructions. We conduct an additional deployment with both the initial agent (with parameters $\theta_1$) and the last deployed agent ($\theta_{11}$). The agents are deployed concurrently in a randomized experiment. Users do not know which agent they are collaborating with. We collect 571 interactions, with a total of 7,784 instructions. Points on the righthand side of Figure 2 (left) show the performance of both models during this deployment. Execution accuracy (estimated with 400 instructions) for the initial model $\theta_1$ improves from an average of 66.7 to 70.4 from user adaptation alone, while the final agent achieves an accuracy rate of 83.6 during this side-by-side comparison. This demonstrates that while roughly 20% of the improvement is due to user adaptation, genuine agent improvement through continual learning is extremely effective.

**User Perception of Agent Behavior** User perception of the agent improves over time. In-game feedback (Figure 2, right) improves: the rate of positive feedback per action increases from 44.9 to 60.3% and the negative feedback rate decreases from 10.1 to 4.7%. The reboot rate per instruction also decreases from 15.2 to 8.7%. After each interaction, we ask the leader for their agreement with several statements. Figure 3 (left) shows the distribution of these post-interaction Likert ratings, which improve significantly over the 11 rounds. For example, the rate of agreement with *The follower completed all the tasks I instructed them to do* increases from 27.7 to 59.8% of interactions.

**Error Analysis** Figure 3 (right) shows trends of seven agent error types across rounds in the long-term experiment. Error types are annotated by workers as part of accuracy estimation. All but one error type decreases significantly over time; the rate of inefficient executions remains stable (Error 6). Manual analysis shows that more than a third of remaining missed-card errors (Error 1) could be attributed to poorly-written instructions, and most extra-card errors (Error 2) occur when the incorrect card shares at least one property with the target, or is closer to the starting position than the target.

**Language Change** Users are likely to change their language over time, as observed in human-human interactions, including in CEREALBAR [11]. In our study, instruction length remains relatively stable over time at an average of 8.6 tokens per utterance. This contrasts with existing findings in reference games, where convention formation facilitates reduction in utterance length over time [24, 9], and shows that interaction dynamics and incentives shape the dynamics of language change. We observe changes in instruction content over time, using heuristics to estimate the number of target cards specified in each instruction. There is a decrease in the rate of instructions that specify no cards to interact with (i.e., cards to select or deselect) from 12.0 in the first round to 7.7% in the last. The rate of instructions specifying a single card increases from 81.5 to 88.7%, while the rate of instructions specifying multiple cards decreases from 6.5 to 3.7%. This is potentially because no-card instructions are relatively inefficient, while multi-card instructions are relatively hard and have a higher risk of cascading errors that require corrections (i.e., deselecting a wrongly selected card). We also find that the rate of instructions containing a reference to an object (e.g., a tree, a pond) decreases over time from 17.9 to 13.5% of instructions. We find that users increasingly shift labor to the agent over time: the average number of steps per set taken by the follower increased from 14.8 to 15.3, while steps taken by the leader decreased from 10.0 to 8.9 steps. These changes illustrate the impact of users developing game strategies and, potentially, a model of the follower's instruction following ability.

### 5.2 Comparison of Learning Design Choices

We conduct a second deployment experiment to study the impact of the amount of initial demonstration data, negative feedback, and reward propagation heuristics. We also compare the feedback learning signal to supervised demonstration data. We design and deploy five system variations:

**REWARDPROP** uses the exactly same process as in the long-term experiment (Section 5.1), including reward propagation (Section 3.2).

**SIMPLEREWARD** measures the effect densifying and de-noising user feedback with heuristics by using only the simple reward (Section 3.2).

**NONEGATIVE** simulates using positive feedback only. We discard all negative feedback, including reboots, from the data, and then apply the reward propagation heuristics (Section 3.2).[11]

**FEWERDEMO** uses only 144 human-human interactions (2,114 instructions) for initialization, about 25% of the supervised data used in REWARDPROP.

**SUPONLY** compares training with human demonstrations vs. feedback data given the same number of interactions. SUPONLY does not use data from human-agent interactions, though we deploy it for evaluation. Instead, after each round, we add a fixed number of additional human-human interactions to the training set, equivalent to the amount of data collected with the other variants. Due to the limited amount of demonstration data, we only deploy this approach for three rounds.

We deploy the five variations for five rounds. In each round, all variations are deployed concurrently, each for 200 user interactions. To ensure that each worker interacts with all agents and to reduce ordering bias, we assign each worker a unique, consistent, and random ordering of agents; as they play multiple games, they cycle through agents in this order. Because all variants except FEWERDEMO are pre-trained on the same initial data, in the first round only, we deploy two agents: one agent for REWARDPROP, SIMPLEREWARD, NONEGATIVE, and SUPONLY, and another for FEWERDEMO. After the first round, we deploy five agents, one for each variation. This experiment cost $16,769.46.

Figure 4 shows results for the five agent variants over five rounds. Overall, learning from feedback is robust to the different design choices. Reward propagation (REWARDPROP) has minor benefit relative to the simple reward (SIMPLEREWARD). REWARDPROP shows slightly faster learning, achieving significantly higher accuracy ($80.3 \pm 1.0$) than SIMPLEREWARD ($78.1 \pm 1.0$) in Round 3, which means SIMPLEREWARD gives a poorer user experience early on, though it catches up quickly.[12]

We observe that negative feedback is important: compared to REWARDPROP and SIMPLEREWARD, NONEGATIVE learns slower, receives less positive feedback, and more negative feedback and reboots.

---

[11]We do not remove the option to provide negative feedback during games in this experiment. Leaders are not informed that negative feedback will be discarded for training. We leave the direct comparison between positive-only and positive-and-negative user interfaces for future work.

[12]Though initial early experiments showed a more striking promise of using heuristics to densify the reward signal, this experiment shows that densification via heuristics is not as influential as initially thought.

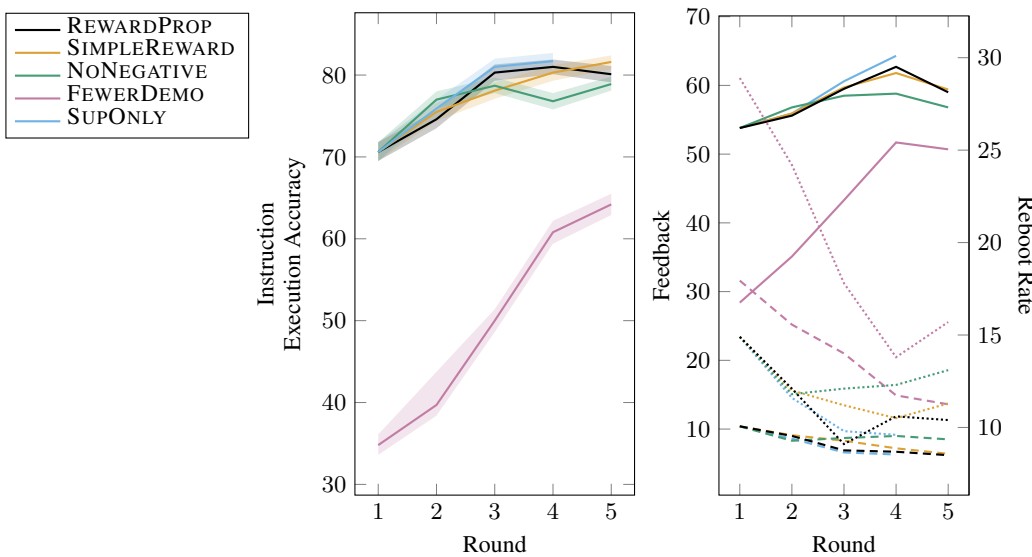

Figure 4: Results for five agent variants over five rounds. *Left*: mean estimated instruction execution accuracy. *Right*: rate of actions with positive (╱) and negative (╱) feedback, and instruction reboot rate (⟳). Across five rounds, all variants exhibit increased execution accuracy, decreased reboot rates, and increasingly positive (and decreasingly negative) action-level feedback.

This difference does not show in early rounds, but appearing around round three. It particularly stands out in round four, when it shows an accuracy of $76.8 \pm 1.0$, compared to $81.0 \pm 0.9$ of REWARDPROP.

The experiment illustrates that learning can start with a much weaker agent. FEWERDEMO uses roughly a quarter of the demonstration data available to the other variants, and indeed starts with much lower performance, $34.8 \pm 1.8$ accuracy, compared to $70.6 \pm 1.1$ for the others. Even this weak model is sufficient for effective learning: it leads to a much steeper learning curve, improving to $64.2 \pm 1.5$ accuracy after five rounds. However, this comes at cost to the initial user experience: in the first round, only 2% of interactions with FEWERDEMO were rated positively by users.

Finally, we observe the feedback data is roughly equivalent to supervised data as a learning signal, showing similar accuracy and feedback trends. REWARDPROP achieves equivalent performance to SUPONLY with less than half the amount of demonstration data (i.e., as used for initialization). This shows the benefits of training through user-agent interaction: not only do learning signals directly target current agent behavior, but the training process is significantly less expensive, requiring no human-human demonstrations, but instead just user-system interactions.

## 6   Related Work

Learning for instruction following commonly relies on various levels of supervision, such as gold-standard demonstrations [e.g., 7, 43, 32, 1, 5, 6, 41, 8, 38], or goal annotations [e.g., 3, 33, 40]. We collapse the distinction between learning and deployment, shifting training and evaluation into human-agent interactions, relying only on a small amount of demonstrations data to initialize.

There has been limited work on continual learning for language-related tasks. A major challenge studying this problem is designing a scenario that allows for scalable non-expert feedback and iteratively improving a model and deploying it. CEREALBAR is unique in supporting real-time interaction with situated natural language between agents and human users. Alternatively, continual learning has been studied via synthetic deployment by simulating feedback using supervised datasets [e.g., 34, 29, 12]. We focus on studies with real human instructors, which create dramatically different dynamics and data cost constraints. For example, this line of work usually uses significant amount of supervised data, while we emphasize rapid learning from limited interactions. Kojima et al. [22] studied the problem of continual learning for instruction generation by observing human following behavior, including deployment in an iterative study. Similar to us, they used CEREALBAR because of

its ability for scalable deployment and real-time model-human interaction. Beyond this environment choice, our work differs significantly: we study human feedback and the task of instruction following.

Human feedback through annotation or selection of intended output has been used for semantic parsing [45, 18]. This is related to soliciting partial annotations [2, 44, 48] or post-interaction feedback [30] in dialogue. Recent work has also employed preference-based user judgments for reward model learning to fine-tune language models using RL [RLHF; 26, 25, 49, 39, 35]. In contrast to this existing work, we focus on sequential execution of instructions in embodied interactions, using realtime feedback provided by human users to continually train our language-understanding agent. While work in this line showed improving a learned model once, it did not study iteratively improving and deploying a model multiple times, a core focus of our work.

Continual learning has been studied more extensively outside of a language context, e.g., for robotics. Our work is inspired by TAMER [20, 21, 46] and COACH [31, 4], where humans provide binary feedback to an agent acting in a world. While these methods were developed in the context of an agent learning a single task, we focus on generalization to previously unseen natural language instructions.

## 7 Discussion

We propose an approach for continually learning to follow instructions through interaction with human users that provide realtime feedback. We demonstrate its effectiveness through multiple rounds of training and deployment, including thousands of interactions. Experiments with various learning design decisions show our approach is robust, and can start from a weak initial agent.

Our work sets the stage for several avenues of further study. First, future work could reconsider a number of our design choices, including the simple contextual bandit objective and the neural architecture. Application of a user simulator [13] could further improve performance; however, this approach fails to capture dynamics that arise in real interactions through user adaptation. An important direction for future work is incorporating additional observational interaction signals (e.g., by observing the instructor behavior) [10, 22]. Such orthogonal signals can accelerate learning. Another promising avenue for research is more expressive feedback mechanisms, such as through natural language and even complete bi-directional dialogue. Although reasoning about such signals significantly complicates both reasoning and learning, they are also much more informative. A potential direction is to use simple feedback signals like ours to bootstrap the language capabilities of the system, and then gradually switch to more complex natural language feedback.

## Acknowledgments

This research was supported by ARO W911NF21-1-0106, NSF under grant No. 1750499, and a gift from Open Philanthropy. Suhr was supported by the NSF under grant No. DGE–2139899, and via a Facebook PhD Fellowship. We thank Noriyuki Kojima for discussions and sharing code, Suyi Diao for contributing to the CEREALBAR implementation, and the participating MTurk workers for their work and help in refining the user experience. We also thank Xiang Lorraine Li and Faeze Brahman, as well as the anonymous reviewers, for providing valuable feedback.

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

# A Limitations and Broader Impacts

Our study was conducted with a group of 108 workers, all recruited from English-majority locales, due to the complexity of recruiting and training workers given the complexity of the task. The group size limits the variation in language use we observe. Its composition restricts our ability to evaluate generalization to other languages, an important direction for future work. Another question for further study is the dynamics created when completely new users join the community in later stages.

Because of the complexity of our studies we kept our architecture close to previous work on CE-REALBAR, and did not study more contemporary architectures or using pre-trained models. We hypothesize using both could lead to better performance, and more consistent improvement trends. This is an important direction for future work.

Our learning setting involves deploying our agent in interaction with human users, which raises time and interaction considerations not present in RL scenarios without a human in-the-loop. This is especially stressed by working in a complex dynamic system where agent observations are coming from a policy-dependent distribution, as users adapt their language and behavior to the previous interactions they have with the agent. Thus, we prioritize sample efficiency when designing our learning algorithm, and opt the simplicity of the contextual bandit optimization problem, which is known to be sample efficient compared to methods maximizing total reward [28, 27]. This also gives us an objective that is mathematically similar to a supervised learning objective, which is a benefit over more complex RL approaches in terms of stability and predictability in learning. Exploring more complex RL algorithms in this setting, e.g., PPO [36], is a compelling direction for future work.

We do not vary the settings of CEREALBAR. Effenberger et al. [11] find that the interaction design and incentives influence the process of language change. Studying the impact of the scenario design decisions would have significantly complicate our experiments, and increase our costs. We decided not to focus on this research question in this work, but treat these parameters as fixed. Although the analysis of Effenberger et al. [11] shows that CEREALBAR creates interesting and complex language dynamics, further study of the impact of interaction design decisions on continual learning is important, and currently under-studied. Our work does not answer these questions, but we hope it will stimulate further research into them.

The data and models we release are not designed to be directly deployed beyond the CEREALBAR scenario. In general, deployment of continually learning systems requires guardrails and monitoring to avoid various undesired outcomes, including acquiring behaviors that may harm users.

# B Model

We implement our policy as a neural network based on the design of Suhr et al. [41]. The inputs are an instruction $\overline{x}$ and observation $o$, and the output is a distribution over actions. The policy architecture is composed of several modules that combine to a single network.

**Embedding Instructions** We embed the instruction $\overline{x} = \langle x_1, \ldots, x_n \rangle$ of length $n$ with a bidirectional recurrent LSTM [15]. This results in a sequence of hidden states $\langle \mathbf{h}_1, \ldots, \mathbf{h}_n \rangle$. The embedding of $\overline{x}$ is the final hidden state of the sequence $\mathbf{h}_n$.

**Embedding Observations** Each agent observation $o$ includes information about the observable environment and the instruction execution so far. The follower agent in CEREALBAR has partial observability. We use a representation similar to that of Suhr et al. [41], but without making the simplifying assumption of full observability. The environment state $\mathbf{W}$ is a tensor representing the properties of each position in the environment as embedding indices. The properties represented in $\mathbf{W}$ also encode information about the follower's trajectory so far, the presence of obstacles in the environment, and the follower's observability. Due to partial observability, each position's representation is derived from its most recent observation; any information that changes about the world may be outdated in $\mathbf{W}$. We embed $\mathbf{W}$ into a dense tensor $\mathbf{W}'$.

**Fusing Embeddings** After independently embedding the instruction and observation into $\mathbf{h}_n$ and $\mathbf{W}'$, we compute a joint representation of both inputs using text-conditioned (i.e., via $\mathbf{h}_n$) convolutions over $\mathbf{W}'$.

**Transforming the Coordinate System** Predicting actions requires interactions between representations of multiple positions. $\mathbf{W}'$ represents the environment using offset coordinates, which do not precisely represent the structure of hexagonal grid in CEREALBAR. We transform $\mathbf{W}'$ to axial coordinates [16], and translate and rotate the tensor such that the center position represents the agent's current location, and the agent is facing in a consistent direction. These transformations are not parameterized.

**LINGUNET** We use LINGUNET [6] to predict the policy distribution over actions $\pi(\cdot \mid \overline{x}, o; \theta)$, with slight modifications to the design of Suhr et al. [41]. For all convolutions, we apply hex-based convolutions with kernels that operate only on voxels within a hex diameter of $d$ around the center voxel, for a kernel size of $d$. We apply instance normalization to the last LINGUNET layer of the input and text-based convolutions. Finally, we do not perform the final transposed convolution. Instead, we directly predict a distribution over the action space given the output of the transposed convolution.

## B.1 Inference

We use ensemble-based inference. Given sets of model parameters $\theta = \langle \theta_1, \ldots, \theta_m \rangle$, we construct a policy $\pi$ over executable actions using voting:[13]

$$\pi(a \mid \overline{x}, o; \theta) \propto \tag{3}$$

$$\exp\left(\sum_{1 \le i \le m} \mathbb{1}_{a = \arg\max \pi(\cdot \mid \overline{x}, o; \theta_i)}\right) \; .$$

Actions are sampled and executed from $\pi(\cdot \mid \overline{x}, o; \theta)$. Executing an action in the environment results in a observation according to the transition function $\mathcal{T}$. We continue to sample actions until the stop action STOP is sampled, or until the leader manually reboots the follower. The STOP action marks the current instruction as complete, which either results in the follower's turn ending, or it receiving the next instruction to follow.

## C Implementation Details

We lowercase and tokenize instructions using BPE [37] with a maximum vocabulary size of 4,096 and a minimum wordtype occurrence of 2.[14] We learn size-64 word embeddings from scratch. We encode instructions with a single-layer LSTM RNN [15] with 128 hidden units. We embed each position's properties into vectors of size 16. We use the same LINGUNET hyperparameters as Suhr et al. [41], and did not perform an additional hyperparameter search.

We use an ensemble size of $m = 10$. We do not train in ensemble, but train ten separate models and apply ensemble-based inference during deployment. When using reward propagation, we use a maximum distance of 8 for propagating to previous actions that received no feedback. For training, we use a batch size of 16 agent steps, a learning rate of 0.001, and ADAM [19] for optimization. We re-initialize model parameters from scratch at the beginning of each round of parameter optimization. We use a held-out subset of the original CEREALBAR training set as a validation set for early stopping, comprising 5% of the original split. After each epoch, we evaluate model performance using SWSD (Appendix D) on the validation set. We use patience for stopping; if ten epochs have passed since the last epoch where the validation SWSD surpassed the previous global maximum, we terminate the training process and choose the model parameters that maximize validation SWSD. We use a single GeForce RTX 2080 Ti for training each model. Training a single model takes about 28.9 hours on average. We run inference on CPU during deployment.

**Comparison of Learning Design Choices** In our second experiment comparing different learning design choices, we deploy for five rounds. This number of rounds was chosen because after five rounds in the long-term experiment (Section 5.1), learning trends were clear; this choice balances

---

[13]We assign zero probability to inexecutable actions, i.e., one that would result in an intersection with an obstacle.

[14]We use the implementation provided by HuggingFace at `https://huggingface.co/docs/tokenizers/`.

experiment cost and insight. If we acquire more than 200 interactions per round because of the crowdsourcing process, we select exactly 200 games for training and analysis by preferring earlier games played by each worker. We discard the other games.

## D  Evaluation

**Instruction Execution Accuracy** For each deployed agent, we randomly sample instruction execution traces $\mathcal{E}_e \subseteq \mathcal{E}_c \subseteq \mathcal{E}$ for manual evaluation. $\mathcal{E}_c$ contains all instructions marked as complete by the agent, and $\mathcal{E}$ contains instructions that were either marked as complete or rebooted.[15] Excluding rebooted instructions from this evaluation creates a biased sample, as reboots nearly always reflect incorrect instruction execution, so we re-adjust accuracy estimates based on reboot rates. We assume all rebooted instructions are incorrect executions. The adjusted correctness rate is:

$$\text{correctness} = \frac{\sum_{\overline{e} \in \mathcal{E}_e} \mathbb{1}_{\text{correct}(\overline{x}, \overline{e})}}{|\,\mathcal{E}_e\,|} \frac{|\,\mathcal{E}_c\,|}{|\,\mathcal{E}\,|} \,,$$

where $\text{correct}(\overline{x}, \overline{e})$ is user judgment of execution $\overline{e} = \langle (o_i, a_i, w_i^a) \rangle_{i=1}^m$ for instruction $\overline{x}$.

**Static Evaluation Data** We also evaluate on the development split from Suhr et al. [41], with *success weighted by stopping distance* (SWSD). SWSD is computed per instruction execution:

$$\text{SWSD}(\overline{e}', \overline{e}^*) = \frac{\mathbb{1}_{\overline{e}'_{-1} \equiv \overline{e}^*_{-1}}}{1 + ||\,\overline{e}'_{-1} - \overline{e}^*_{-1}\,||} \,,$$

where $\overline{e}'$ is the trace of the agent's execution of an instruction and $\overline{e}^*$ is the human demonstration. $\overline{e}'_{-1} \equiv \overline{e}^*_{-1}$ only if $\overline{e}'$ results in the same set of cards selected as in $\overline{e}^*$. $||\,\overline{e}'_{-1} - \overline{e}^*_{-1}\,||$ is the hex distance between stopping positions. SWSD is stricter than simple card-state accuracy [41], as it gives only partial credit to instructions where an execution stops in an incorrect position.

## E  Crowdsourcing and Data Details

This study received an exemption from the Institutional Review Board of the institution where the research was conducted. Worker identities are anonymized in the data we release.

We qualify workers through a tutorial and a short quiz about the game rules. Workers are also required to reside in an English-majority locale and have a HIT approval rate of over 90% with at least 100 approved HITs during their time on MTurk. The base pay for completing the qualification task is $0.50 USD, and qualified workers receive a $2.00 bonus. We qualify 108 workers. Following Suhr et al. [41], we pay workers bonuses per point earned in each game, increasing the compensation per point as the game score increases. On average across all experiments, each game costs $2.91 and workers are paid an average of $21.85 per hour.

We split workers into two pools: expert and novice. Expert workers earn a 50% higher bonus per game than novice workers. Workers are moved to the expert pool after playing at least two games with a score greater than zero, as long as their rate of giving feedback is greater than 75% of instructions.[16] Workers return to the novice pool if they play for two rounds with a feedback rate of less than 75% of instructions. 65 workers achieve and maintain expert status throughout the experiments. Only expert workers are qualified to provide post-hoc instruction execution judgments, where they are paid $0.07 per judgment of instruction execution. Worker IDs are anonymized in the distribution of interaction data. Figure 5 shows the instructions provided to workers in MTurk, and Figure 6 shows the CEREALBAR interface during interaction.

**Agreement** For about 20% of instruction execution receiving post-hoc judgments, we acquire judgments from three workers; we find that for only 3.6% of these no consensus is achieved among the three workers, which indicates very high overall agreement.

---

[15]In this evaluation, we ignore all instructions that were not completed due to the game ending.

[16]The rate of feedback per instruction measures the proportion of instructions where at least one action in the follower's instruction execution is given positive or negative feedback, including a reboot.

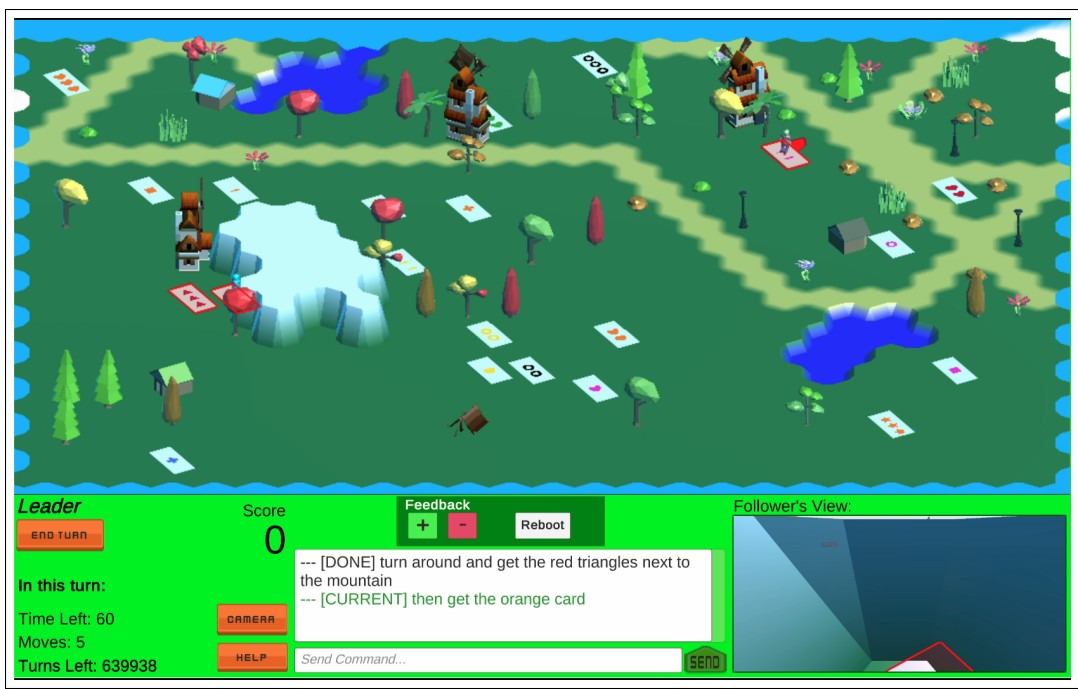

**In this HIT, you'll play a collaborative game with a partner.**

- You will be a *Leader* and will work with an automated *Follower* (i.e., not another worker).
- You should be able to connect quickly to your partner. If you spend more than a minute or two waiting for a partner, please contact us.
- If you are confused about the game, please read the instructions on the game page, or visit this page. You can also email us at [REDACTED FOR SUBMISSION].

Check the HIT title to see whether you are in the *novice* or *expert* group.

HITs in the expert pool have about a 50% increase in bonus amount over to the novice pool. See the instructions page for details and the bonus scheme for novice and expert workers.

We will evaluate workers in the expert batch periodically to make sure they are still adhering to our expert standards, and we will evaluate novice workers after every batch to see if they can move into the expert pool.

- Some things we will take into account for the Leader games:
  - You should move around the map with your partner to collect sets.
  - Your commands should be unambiguous and easy for your partner to follow.
  - You should understand what makes a valid set (three cards with different colors, shapes, and numbers).
  - You should be actively giving good/bad feedback to your partner (at least a few signals per instruction).
  - You shouldn't use the Reboot button unless the Follower is performing poorly.
  - **Give feedback for over 75% of instructions (feedback can include rebooting the follower).**
- To get into the expert pool, novices need to complete at least 2 games with a score of at least 1, and give feedback for at least 75% of instructions.

**Please report any issues you see in the feedback box below, and take screenshots where possible (email to [REDACTED FOR SUBMISSION]).**

**Don't attempt to play more than one game at a time.** Our server will only accept a single connection from each worker at a time. We appreciate your full attention while playing the game!

Figure 5: Instructions provided to workers for the main task on MTurk. Detailed instructions about gameplay were provided on a separate set of webpages, and will be available alongside our code when released.

Figure 6: The CEREALBAR interaction interface. Users provide instructions in the command box, and feedback via either buttons in the GUI or keypresses. The follower's partial view of the environment is visible in the bottom righthand corner of the interface.

# F   Additional Results

## F.1   User Perception of Agents for Comparison of Learning Design Choices

Figure 7 shows the Likert distribution for the three post-interaction statements users are asked about for the experiments comparing learning design choices, where we concurrently deployed five systems for five rounds.

## F.2 Evaluation on Static Data

Evaluation through human-agent interaction is the main focus of our work. However, we also evaluate instruction-following agents against held-out, static data from Suhr et al. [41]. This evaluation does not take into account how the actual data distribution shifts over the agent's lifetime, because of the dynamics between the agent and human users. Figure 8 shows average SWSD for the models deployed in each round. SWSD begins at 39.7 for the initial model, and peaks at 46.4. This improvement is due entirely to adding training data acquired from human-agent interactions.

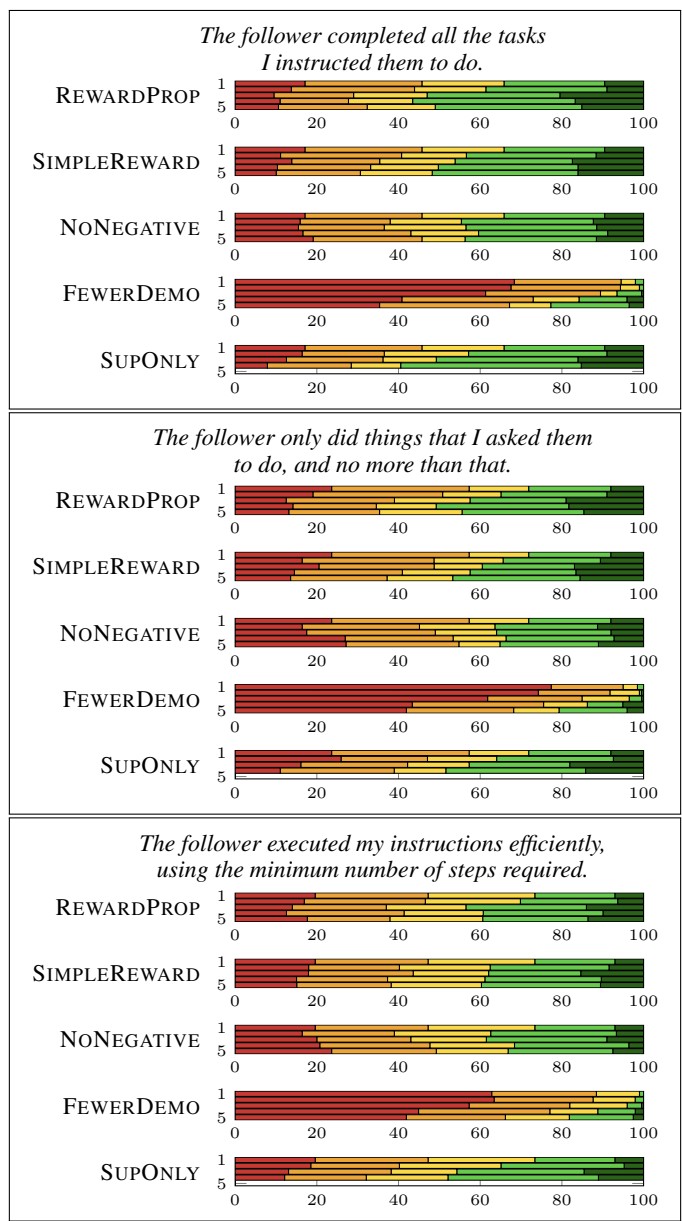

Figure 7: Distribution of post-interaction user agreement with three statements about the follower's performance for our approach comparison experiment.

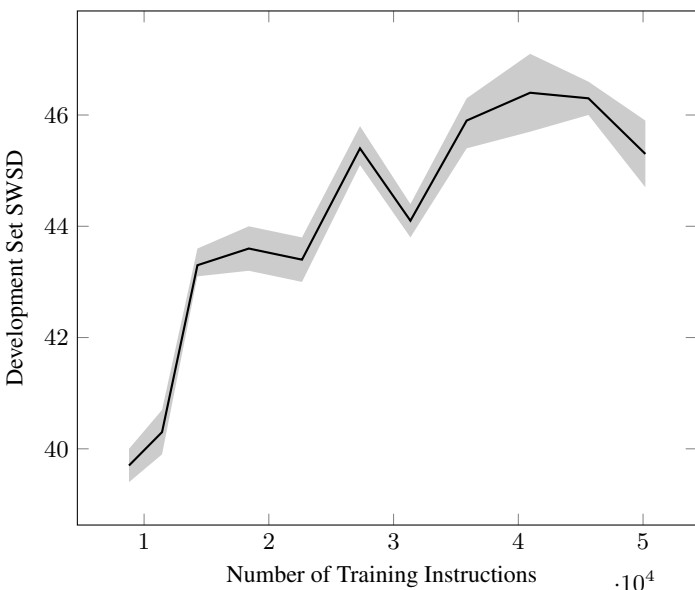

Figure 8: SWSD on the held-out development data, averaged over five runs of sampling-based inference.

