# OpenReview forum: "Continual Learning for Instruction Following from Realtime Feedback"
_NeurIPS.cc/2023/Conference — NeurIPS 2023 spotlight_

### Official Review · Reviewer_4ZhD · 2023-06-27

**Soundness:** 3 good
**Presentation:** 3 good
**Contribution:** 2 fair
**Rating:** 5
**Confidence:** 2

**Summary:**

Training of a continuously learning, instruction following agent from feedback provided by users during collaborative interactions. The problem is that humans often give noisy reward and at irregular intervals. The method formulates the learning scenario as a contextual bandit problem and alternates between training and deployment stages. The main issues addressed is the irregular timed reward and credit assignment problem. The method addresses this using heuristics and demonstrated its effectiveness in the CREALBAR environment using human evaluation.

**Strengths:**

The paper is straightforward to follow. The topic is very relevant. The evaluation and improvement throughout the interaction rounds are significant in many aspects. The experiments conducted are very thorough and complete and have convincing results to highlight the continual learning nature of the agent.

**Weaknesses:**

- The assumption of feedback is matched with an action or close by actions (heuristics presented in the paper) is somewhat restrictive
- transitions without rewards are discarded and not fully utilised, quite a waste of resources
- requires expensive labelling process

**Questions:**

- if I understand correctly, since the method formalizes itself as a contextual bandit problem, maximizing the immediate expected reward, Line 77 says "Feedback signals are given in realtime, and are not timed to coincide with a specific action" will not be favorable for training since the corresponding action and reward is not even matching up?
- isn't the assumption of giving a binary reward and optimizing for the contextual bandit objective amounts to gradient descent/ascent on the right/wrong actions (with noise, since humans give reward quite randomly), which is almost equivalent to supervised learning? If this is the case, I feel like the continual learning part is just iteratively collecting more supervised learning data, which performance improvement is quite obvious


**Limitations:**

Limitations and broader impacts are complete and included in the appendix

---

> ### Author Rebuttal · Authors · 2023-08-09
>
> We thank the reviewer for their comments and questions. We are looking forward to answering any follow up questions during the discussion period.
>
> Assumption of feedback alignment: real time feedback follows patterns of human response, including its delays. So this assumption follows the role of humans in the process and how interactions happen in real time. It’s possible for humans to give feedback to decisions that were made further in the past. Our learning approach does not handle it, and will suffer when such a signal is given. However, this is not something we force our users to avoid, and empirically the approach handles this and other types of noise quite well, showing effective learning. In addition, this correspondence between feedback and action is also seen in prior work (e.g., COACH and TAMER). We agree it does not provide perfect credit assignment (an important direction for future work), but our approach is robust to it. Our heuristics make further assumptions, but our experiments generally show that they are not critical for effective learning.
>
> Discarded training data: there are two conditions. When we don’t use heuristics, the data is essentially discarded. When applying heuristics, nearly all actions/transitions are assigned feedback (and thus used during training), except those that appear after the last feedback provided by a user in a rollout, as we can’t estimate what feedback the user would have given. This was a key motivation behind the heuristics. However, the heuristics didn’t prove critical, as noted above. It remains an open problem for future work to study if it’s possible to infer a learning signal for actions without feedback. We don’t claim to completely solve the problem of learning from such feedback, but to show an effective approach and its significant potential.
>
> Expense of labeling: except the supervised data used for initialization (a very small amount) and the human evaluation, there is no conventional labeling cost involved in the actual continual learning. The cost of our experiments is paying for people to interact with our system, and user feedback is built into the interaction. It’s just like deploying a system, but we don’t have the ability to deploy a real product for people to interact with, so we create this scenario by paying people to act as users. This approach is advantageous not only against conventional data annotation, but also when compared to most RLHF methods, which rely on post-hoc preferences from third-party annotators (and then you do see high labeling costs).
>
> Questions:
>
> Q1, alignment between actions and rewards: To clarify this sentence, we mean that we don’t force users to provide feedback for each individual action the agent takes (e.g., by moving the agent only after the user gives feedback for the current action), but we instead allow users to provide feedback at any point in time as they observe the instruction being executed. This creates a more challenging signal than forcing humans to provide feedback after action (which would give perfect alignment), but would make for a very frustrating system use. In practice, alignments between the user-provided feedback and the intended actions are very good, especially after correcting for reaction time delays (as is common when handling human responses). There’s still noise, and the credit assignment problem is not completely solved. But our approach is robust to it, as demonstrated by our experiments that show effective learning over time.
>
> Q2, equivalence to supervised learning: the objective does look a bit like supervised learning (except the IPS coefficient). However, it only appears like a supervised learning objective: the actual supervision isn’t coming from a gold-standard demonstration (i.e. a human demonstration of instruction execution). Instead, the data we train on includes trajectories sampled from our policy (conditioned on user instructions) annotated with feedback provided by users. Additionally, unlike supervised scenarios (and single-agent RL scenarios), this is a non-stationary environment, because humans constantly change their behavior. So the data is not coming from the same constant distribution, as assumed in supervised learning. So in practice while the objective appears similar to a supervised learning objective, the learning problem is far from a supervised learning problem. The objective is also derived from maximizing the value function (i.e., like in REINFORCE), and not by maximizing the data likelihood. But, the mathematical similarity to supervised learning is actually a benefit of our method, as supervised learning is more stable and predictable than many more complex RL approaches.
>
> Another small note: while user-provided feedback is somewhat noisy, it is not so random that it is impossible to learn from, and analysis of the human-provided feedback shows that it is very high quality. This is also demonstrated by our results, showing very effective learning.

---

> > ### Comment · Reviewer_4ZhD · 2023-08-17
> >
> > Thank you for your clarifications. Based on a better understanding of how alignment is achieved through continuous interactions between human/agent, I updated my score to 5.

---

### Official Review · Reviewer_Q7Gw · 2023-07-07

**Soundness:** 3 good
**Presentation:** 3 good
**Contribution:** 3 good
**Rating:** 7
**Confidence:** 3

**Summary:**

The authors propose a method for online continually training an instruction-following agent based on user realtime feedback gathered for a collaborative game CerealBAR. The agent need to follow the human's  instructions and complete the task. The paper utilizes the contextual bandit learning approach, with  immediate rewards computed from user binary feedback. Through the evaluation of thousands of human-agent interactions, they demonstrate a significant improvement in instruction execution accuracy over time. They also conduct experiments with multiple training scenarios and verify the robustness of their approach.

**Strengths:**

* The authors propose and deploy the methods in the collaborative human game. They collected agent-human interaction data will be helpful for future agent design.
* It is a pilot work to demonstrate the strength of continual learning from human feedback for instruction-follow agent. The authors shows that after training the agent from human-human dataset, the policy can be greatly improved from the online off-policy learning from human binary feedback.
* The authors conduct comprehensive online experiment with variants of learning choice and human post-interaction evaluation. The variants of learning choice show the robustness of the framework and the human post-interaction evaluation shows empirical analysis on the agent behavior improved after learning from human.

**Weaknesses:**

* The framework deign is not very clear. The authors do not mention much the design of the policy model, which transform from human instruction and observation into action (Model part in section 4). This paper mainly discusses about how to define user feedback and how to train with continual learning given the feedback.
* The framework uses binary signal from the user feedback, which is limited to represent the human satisfaction or feedback. It can be difficult to generalize to other tasks with complex instructions and observations.
* The framework is relatively simple with REINFORCE algorithm. It would be more convincing if the authors can conduct more experiments on other standard off-policy learning algorithm for the robustness of the framework.

**Questions:**

Although the model can achieve 15.4% absolute improvement through continual learning over time from continual learning, it is hard to gain insight of how good the an instruction-following agent is. In order to show benefit of off-policy learning in this framework, have we tried some baseline models for comparison, such as offline RL, due to the difficulty of additional online data collection?

**Limitations:**

The experiments is difficult to reproduce. Training policy with from human realtime feedback can lead to significantly higher cost compared with offline RL or user simulator, which is widely used in dialogue agent design.

---

> ### Author Rebuttal · Authors · 2023-08-09
>
> We thank the reviewer for their comments and questions. We are looking forward to answering any follow up questions during the discussion period.
>
> Model design: We provide the design of the policy model in the Appendix. In short, it is a modification of the architecture used by previous work (Suhr et al. 2019). As it is not a main contribution of our work, we have put it in the appendix. Our aim was to take a model that has been shown to work to some extent on this domain, and focus on the learning problem.
>
> Binary feedback: Existing work on learning from human feedback for embodied agents uses this kind of binary, realtime feedback provided by users (e.g., in TAMER and COACH). Binary feedback is simple, and conceptually generalizes to other tasks and feedback scenarios. Effectiveness is an empirical question, of course. However, for example, LLM RLHF uses binary preference signals (albeit unlike our work: given by third party annotators, and are not embodied, not realtime), and which empirically works well for complex reasoning tasks. The simplicity of binary feedback is what allows interleaving it with the interaction. It also doesn’t require complex machinery to interpret it (like natural language feedback). Of course, it carries less information than other forms of feedback (e.g., natural language), even if it requires less of the user. Using such information rich (but more costly to obtain) forms of feedback is an orthogonal and open research question. They come with pluses (more information) and minuses (costly to get, harder to interpret).
>
> Re experimenting with off-policy algorithms: please see the general response and “on using offline RL” below.
>
> Insights on policy quality: We provide comprehensive analysis of policy errors as annotated by workers in Figure 3 (right), which we discuss in the main paper. We see that for nearly all of the error categories, the policy is improving over time; the only error category which remains roughly stable is the efficiency of the policy’s trajectory (roughly 6% of trajectories are marked as inefficient).
>
> On using offline RL: we agree there are alternative offline RL methods that we could use to swap our REINFORCE-style optimization loop. Integrating them in this way won’t reduce the need to collect feedback data over rounds. These methods are also more complex, while we opted for simplicity in this first study of the problem (there are many open directions for future work). Swapping the whole process with offline RL is less clearly effective. We could swap the supervised initialization using offline RL with the seed data, and that might increase initial performance, or allow us to use less initial data. This would just give a better launching pad for our learning from feedback. Of course, this is an empirical question, which is orthogonal to the questions we study.
>
> On using a user simulator: While using a simulator could potentially improve model performance as it would generate new data to train on, it fails to capture a fundamental aspect of agents deployed in real interaction with people: the dynamics that arise as real users adapt to the agent through interaction. Manually designing user simulators has been studied in the past and is orthogonal to our contributions; such a manually designed simulator would also be placing assumptions on the kinds of language that people use in interactions with our agents. Building a strong simulator is also a lot of engineering work. Our approach takes advantage of the interaction the system would have with users anyway, and because it derives data from these interactions, it’s always aligned with the data distribution observed in its actual interactions, something that is very difficult to achieve with simulation.
>
> Reproducibility: Experiments with real human users are naturally more difficult to reproduce than experiments which rely on fixed / static datasets or simulators; we anticipate that all of the difficulty in reproducibility will arise from recruiting and maintaining a user base. However, we publicly provide code, implementation details, data, etc. Furthermore, conducting these experiments at the scale we did is a key contribution of our paper. Almost no open research that is reported in detail has been done with such deployments over long periods of time, as we do. The complexities are a natural by-product of the core research problem we study, and there is no way around them. Simulators, for example, avoid costs and increase reproducibility, but poorly reflect interactions with real users. For example, simulators are static, whereas real users are dynamic and constantly change their behavior. On a fundamental level, there’s a big difference between the stationary problem simulators provide, and the non stationary problem of a deployed system, as we study.

---

> > ### Comment · Reviewer_Q7Gw · 2023-08-18
> >
> > Thank you for elaborating! The overall design is reasonable and the online continually learning study is valuable for future research. I updated my score to 7.

---

### Official Review · Reviewer_d6RP · 2023-07-07

**Soundness:** 3 good
**Presentation:** 4 excellent
**Contribution:** 3 good
**Rating:** 7
**Confidence:** 3

**Summary:**

This work demonstrated a simple yet effective framework for continual learning in instruction following task utilizing human feedback. Using CEREALBAR as testbed, this work demonstrated the framework in abundant details, and show effectiveness through experiment results. This work also conducts various analysis related to human feedback during and post-hoc the interactions. Finally, various decision choices are compared.

**Strengths:**

The method is new and well supported by the framework description, the experiment results, and the analysis in the paper. The writing is well organized and clearly written in general.

This work address embodied ai problem from a relatively new perspective. It demonstrates effectiveness on task performance as well as benefits on reducing annotation effort. This work also provides promising future directions centering around the human-agent dynamics like the form of human feedback, the communication style between human-agent.

**Weaknesses:**

I don't find any noticeable weakness of this paper.

**Questions:**

1. Figure 3: left, the x axis range indicates number of turk workers right? aren't they 108 in total?
2. Maybe I missed something, but how do you annotated the demonstration data? Also, since reward propagation is one of the main design choice, does the gold data's feedback distribution similar as those annotated by mturkers?
3. Do you have any clue on how user adaptation contribute to specific error decreasing? (figure 3, right)
4. line 271: not sure what "we deploy a single agent for all but FEWERDEMO in the first round" means. Can you explain more?
5. (Figure 4, right)There seems to be a golden ratio between positive and negative feedback emerging. What do you think of that?

**Limitations:**

Limitations are adequately addressed in supplementary material.

---

> ### Author Rebuttal · Authors · 2023-08-09
>
> We thank the reviewer for their comments and questions. We are looking forward to answering any follow up questions during the discussion period.
>
> Q1, Figure 3 x axis: The x-axis represents a proportion of interactions (so it sums to 100), rather than the exact number of interactions. Each different colored bar represents the proportion of interactions in each round that were given a particular rating. We are using proportions here because the number of interactions per round differs slightly. We will clarify this in the paper.
>
> Q2, demonstration data: To “annotate” the supervised demonstration data with feedback, we assign every action positive feedback, because we assume human demonstrations are gold-standard (i.e., everything a human follower did in the demonstration is correct). We don’t apply reward propagation heuristics to this data, because we assign positive feedback to each action, so in terms of the feedback distribution, demonstration data includes 100% positive feedback while the data collected during human-agent interaction includes examples with negative feedback.
>
> Q3, adaptation and errors: This is a great question. We didn’t find that any particular errors correlated with particular user adaptations. We find that in the direct comparison between the initial and final models ($\theta_1$ versus $\theta_{11}$), where user adaptation is reflected in interactions (when both are deployed concurrently in the final round), the initial model still has significantly more errors in all categories except for Errors 5 (which is already very rare) and 6 (which stays static over time anyway). This comparison further shows the error trends reflect genuine improvements in the model, and are only partially influenced by user adaptation (although we indeed see some adaptation, and our experiment measured for it).
>
> Q4, deployment: Thanks for bringing up this confusing wording. What we mean is that for {RewardProp, SimpleReward, NoNegative, SupOnly}, the agent in the first round is exactly the same, as all of these systems are pre-trained on the exact same set of training data, so there’s no difference between them in the first round. But, for {FewerDemo}, the model is pre-trained on a smaller subset of the data. So in practice in the first round, we only deploy two agents: one for {RewardProp, SimpleReward, NoNegative, SupOnly} and one for {FewerDemo}. We will clarify this in the paper. This allowed us to save some interactions (and money) without influencing the experiment.
>
> Q5, feedback ratio: this is an interesting question, and we don’t have a conclusive answer to it. It does seem like the curves converge to a similar ratio across the different systems. We suspect we would need to run this experiment for much longer to confidently say that a consistent ratio emerges. This would likely require re-running with a disjoint pool of workers as well, to rule out the impact of the specific pool of workers. So, we don’t think we are ready to draw strong conclusions, but we do see what the reviewer is pointing to. This raises interesting directions for followup work, which our work enables, such as the impact of learning and interface design decisions on the long-term equilibrium of human user feedback behavior.

---

> > ### Comment · Reviewer_d6RP · 2023-08-17
> >
> > Thanks for addressing my questions! It is a very interesting work and the effort you put in the work is respectable. I will update my score to 8

---

### Official Review · Reviewer_hEYa · 2023-07-10

**Soundness:** 4 excellent
**Presentation:** 4 excellent
**Contribution:** 3 good
**Rating:** 8
**Confidence:** 5

**Summary:**

This strong work presents a systems contribution in a fully-fledged system for continual learning from language feedback, in the context of situated human-to-robot instruction following tasks. Using the CerealBar environment (roughly inspired the card game SET, with an embodied flair), this work starts by learning an agent's instruction following policy from a set of offline demonstration data. This agent is then deployed against a crowd of *real human users* who work together with the agent to complete more and more task instances in the CerealBar environment, with humans providing (sporadic) binary feedback in real-time.

After each learning round (a fixed number of episodes), the agent policy is updated from the real-time reward data, where the authors define a heuristic to assign the real-time feedback to individual (state, action) pairs taken by the robot; this heuristic is derived from empirical "delay in human response" data, as well as from the traditional RL literature for assigning credit to actions that aren't explicitly provided feedback. The agent update is a contextual bandit style update, using a variation of the simple REINFORCE policy gradient update (training effectively a 1-step RL policy).

Across 11 rounds of updates from real-world human data, the proposed system obtains impressive results, showing a constantly growing trend in both individual instruction execution accuracy, as well as total CerealBar score (with the steepest increase in performance happening across the earlier rounds). The work also has a series of systematic ablations (comparing the binary feedback based approach with "full supervision" - showing that the proposed bandit approach can almost match the same performance), as well as qualitative Likert scale results from the real crowd users actually interacting with this system.

**Strengths:**

This is a well-written and extremely well-executed systems contribution - the task of learning to improve situated agents from language and binary feedback is incredibly timely, and every part of the proposed system is implemented carefully and studied thoroughly. The evaluation is strong (both qualitative and quantiative), and definitely show that not only is the agent able to effectively improve with feedback, but it's able to improve *efficientyly* and actually grow its capabilities over time. I am a huge fan of this work.

Separately, I think the dedication of this paper to open-sourcing all parts of the pipeline, including the multi-round crowdsourced interaction data is admirable, and will be an incredible resource to the community.

**Weaknesses:**

In a paper such as this, I can understand it's very hard (costly + time-consuming + introduces a lot of confounds) to evaluate multiple different learning approaches at the full-scale of the real-world evaluation (though I really appreciate the experiments in section 5.2 that does this for a limited number of rounds). That being said, I would love to look at other agent learning paradigms, and justify the use of the 1-step contextual bandit style reward, vs. a multi-step RL approach that automatically learns how to perform credit assignment given the sporadic binary feedback. Especially using some of the more recent tools in the RL toolbox (e.g., learning a value function/advantage function, PPO-style clipped updates, or even off-policy/offline RL methods).

A general worry I have is that much of this paper hinges off the design of the reward/utility function, which starts from the principled binary feedback provided by users, but is then further processed through a series of heuristics that may or may not really capture what's going on (for example, how the current heuristic labels transitions when there isn't an explicit binary reward tied to that timestep). I would love for the authors to address these choices in a bit more detail.

**Questions:**

What motivated the choice of using clipped IPS coefficient for the policy gradient update vs. a PPO-style update (with a learned one-step value function estimator for debiasing); given the choice of using REINFORCE policy gradient anyway for the 1-step update, this would've been an easy addition, and possibly more stable?

I'd be curious to see a more detailed breakdown of the types of language instructions (and *when* in an episode a user usually provides feedback) across the different rounds; are there more complex instructions/abstractions learned over time? Do users intervene differently in different rounds? What are the failure modes?


**Limitations:**

The paper provides a thorough limitations section (in the appendix) that is well thought out and clearly states weaknesses in the current approach. It would be nice to move some of the punchlines to the main body in the final version of the paper though!

---

> ### Author Rebuttal · Authors · 2023-08-09
>
> We thank the reviewer for their comments and questions. We are looking forward to answering any follow up questions during the discussion period.
>
> Other methods: We agree that experimenting with different learning methods is a good direction for future work; in this case, we opted for simplicity especially in light of only having a small amount of data to learn from (methods which require learning multiple models, such as a value function, may have stronger requirements in the amount of data used for training), and working in a complex dynamic process where agent observations are coming from a policy-dependent distribution (i.e., because users adapt their language and behavior to their interactions with the agent). The paper also explores uncharted territory, so we figured simplicity is the best way to approach it, even if higher performance is likely with more complex methods. As the reviewer notes, cost was also a factor, as more complex methods often require more tuning. We hope our work paves the way for future work in this area, including along the directions the reviewer mentions.
>
> Heuristics: We manually developed heuristics initially after early experiments showed they were useful. In general, the rate of “errors” assigned by the heuristics was very low (i.e., nearly always reflected the instruction’s intent), so heuristics serve to densify reward in our contextual bandit setup. Surprisingly to us, in our comparison experiments (Section 5.2) we found they were actually not as influential as initial pilot experiments showed; however, there is some evidence they sped up learning early on. So, an important takeaway is that the effectiveness of the approach does not necessarily rely on very involved heuristic reward design (although future work might reveal better approaches that do make significant difference).
>
> Policy gradient vs. PPO-style learning: see above under “other methods”. This was motivated by preference to simplicity, especially given costs of studies and the goal of rapid updates with a very limited amount of data. Early on, we conducted pilot studies with more complex methods (e.g., COACH by MacGlashan et al. 2017, which is designed to learn from feedback), but they didn’t work well. That said, there were too many confounding factors to draw strong conclusions about the algorithm, except that getting them to (potentially) work is more complex. An interesting direction for future work now that we showed the whole process works is to apply more localized changes to the optimization algorithm (e.g., switching to PPO as you suggest) and understanding the implications. The rationale for using IPS is (a) to avoid exploding gradients and (b) de-bias (end of Section 5).
>
> Analysis: We didn’t find evidence that users built abstractions or relied on more complex instructions over time; in fact, there is some evidence that they simplified their instructions (e.g., decreasing the rate of multi-card instructions), and reducing the number of references to objects in the environment. We don’t think the CerealBar stimuli is designed to elicit abstractions. In terms of interventions, we found that the rate of user intervention via reboots goes down significantly over time (as the agent gets better). Also, while this is not reported in the paper, we found that users shifted more labor to the agent over time: the average number of steps the follower took per set increased from 14.8 to 15.3 over the long-term experiment, while the number of steps for the leader decreased from 10.0 to 8.9 steps. We can add this analysis to the main paper. If by failure modes you are referring to common errors in instruction execution, Figure 3 (right) reports such categories and their trends via manual analysis, with discussion in Section 5.1.
>
> Thank you for the suggestion to move limitation highlights to the main text! We will certainly do this if the paper is accepted.

---

> > ### Comment · Reviewer_hEYa · 2023-08-16
> > **Rebuttal Response**
> >
> > Thanks for responding to my review! I think this is a very strong paper, and highly encourage the other reviewers to engage with the authors and consider raising their scores as well!

---

### Official Review · Reviewer_zvAL · 2023-07-21

**Soundness:** 2 fair
**Presentation:** 2 fair
**Contribution:** 2 fair
**Rating:** 5
**Confidence:** 3

**Summary:**

The authors work on the CEREALBAR setting, where two agents (one human and one computer) cooperate using natural language to achieve a shared goal. Specifically, the authors propose a new setting where the human agent can provide binary feedback to the computer agent. In their new setting, the authors follow the contextual bandit setting and use a REINFORCE-like algorithm to train the agent. Overall, experiments show that the agent achieved an improvement by learning from human signals.

**Strengths:**

1. The authors explore a setting that involves real-time human feedback, which may be under-explored.
2. The experiments involve significant human labour and may be beneficial to the community if the authors share their data.

**Weaknesses:**

1. Limited contribution. The authors simply applied existing algorithms to their settings.
2. Experiments show little insight. Their experiments demonstrate that the agent improves by receiving more human feedback, which is expected.
3. (Minor) Lack of concrete numbers. Results are mostly shown in figures, and it's difficult to see the real performance numbers.

**Questions:**

1. Why do you use a contextual bandit rather than the full RL setting? This is a bit unintuitive since you have a sequential decision-making process.
2. In Figure 3, it seems the performance is the highest at around round 5. Is there a reason for this?
3. Do you plan to release the collected human feedback data? I checked the data folder in the supplementary file but didn't see them.
4. What models are you using? The paper mentions neural networks, but could you please give more details?

**Limitations:**

Limitations are discussed in the Appendix.
For example, the authors openly disclose that they did not use a more modern architecture for their agent.

---

> ### Author Rebuttal · Authors · 2023-08-09
>
> We thank the reviewer for their comments and questions. We are looking forward to answering any follow up questions during the discussion period.
>
> Contributions: As the reviewer suggests, indeed, learning from real-time human feedback in embodied interactions is certainly an underexplored area. There is a small amount of existing work in robotics that used real-time feedback to train non-language policies, from both explicit (e.g., TAMER – Knox et al. 2009, COACH – MacGlashan et al. 2017) and implicit (e.g., EMPATHIC – Cui et al. 2020) human feedback. But all this is not focused on natural language. Reinforcement learning from human feedback (RLHF, as in Ziegler et al. 2019 and more recently) is focused on preference-based feedback (i.e., comparing two possible outputs) provided by annotators outside of the interaction context; this work is also not embodied. While our work builds on existing techniques, our learning process as a whole, including experimenting with different ways of mapping feedback to rewards, is a new contribution. The objective we use is a variation of the well known policy gradient REINFORCE objective (although we use it in a bandit setting), but it's part of a complete learning process that is our contribution.
>
> Data sharing: We will share all our data publicly under the MIT license for maximum usability (briefly noted in the paper). The data is included in our supplementary material, in the `game_recordings/` directory; the `data/` directory includes classes and utilities for managing the data. We apologize for the confusion.
>
> Numbers in figures: We specify key numbers in the text. We don’t discuss concrete numbers from Figure 3 (left) or Figure 4 (right) and only highlight general trends in the text due to space limitations. The figures try to balance readability of trends with accuracy. We will add concrete discussion of these two subfigures into the main text, with the exact key numbers. We will also release the complete raw data underlying each of the plots.
>
> Questions:
>
> Q1, contextual bandit (CB) objective: Generally, there's a significant credit assignment challenge in a sequential decision problem like this one, and naively computing discounted returns can lead to many wrong reward assignments. CB avoids this, even if at the expressivity cost of restricting the reward to a single step. Studying better solutions to the credit assignment problem is an important problem for future work. Note that most recent RLHF work also uses a contextual bandit reward, albeit even more restricted because reward is assigned to the complete output (even though text generation can be cast as a sequential decision process). So we are not the only ones opting for this simplification. Also, theoretically, CB has sample complexity advantages with much tighter sample complexity bounds when comparing upper bounds for contextual bandits (Langford and Zhang, 2007), even with an adversarial sequence of contexts (Auer et al., 2002), to lower bounds (Krishnamurthy et al., 2016) or upper bounds (Kearns et al., 1999) for total reward maximization. This is critical in our scenario where we learn rapidly from few examples.
>
> Q2, performance in Figure 3: Figure 3 shows likert scores for post-interaction user questions and error categories, so we are unsure of where this conclusion is being drawn from – could you please clarify where in Figure 3 there’s evidence that performance is highest around Round 5?
>
> Q3, feedback data release: Yes, please see above. The data is included in the `game_recordings/` directory of the supplementary. It will be released under MIT license.
>
> Q4, models: We use a model based on prior work (Suhr et al. 2019), where the inputs (instruction and observation) are embedded and a modified LingUNet (Blukis et al. 2018) is used to predict an action. This is specified in Section 4, and details of the model architecture are available in the Supplementary Material.

---

> > ### Comment · Reviewer_zvAL · 2023-08-16
> > **Reply to the author**
> >
> > Thank you for the response. I really appreciate it!
> >
> > As for contribution, I think the dataset can be beneficial for the community. However, I feel more analysis can be really helpful if the entire learning process is to be claimed as a major contribution. Currently, the experiments show that more human data improves performance, but I think that is not surprising and doesn't fully justify the approach. I feel more baseline approaches can help improve the paper.
> >
> > My question was about the left-hand side of Figure 3, where "strongly agree" usually peaks at around round 5.

---

> > > ### Author Response · Authors · 2023-08-17
> > >
> > > We conduct analysis via deployment experiments comparing variants of our approach (finding it robust to several modifications, that using negative feedback helps, and that we can roughly match performance of a learning approach that requires significantly more expensive training data), analysis of how user adaptation occurs and the extent of its influence on perceived model performance, as well as analysis of model errors and how they change over time through continual learning. We are happy to add additional analyses that the reviewers consider as important to demonstrate the effectiveness of our approach, and provide further insight into our experiments.
> > >
> > > With respect to Figure 3, the broader trend of agreement with the statements (agree + strongly agree) are increasing consistently over the rounds (likewise, rates of disagreement are also consistently decreasing over time). It’s critical to consider the full range of ratings for a complete view of trends.

---

> > > > ### Comment · Area_Chair_SdoR · 2023-08-17
> > > >
> > > > Thank you for the rebuttal and discussions!
> > > >
> > > > To Reviewer zvAL:
> > > > It seems the remaining concerns are regarding more analysis and more baselines. Can you please elaborate? Thanks.
> > > >
> > > > AC

---

> > > > > ### Comment · Reviewer_zvAL · 2023-08-17
> > > > > **Reply to the Authors**
> > > > >
> > > > > I am happy to provide suggestions. These experiments may contain flaws, but I hope they can convey my concerns.
> > > > >
> > > > > 1. Using offline human-agent interactions to provide a performance upper bound. Currently, while the performance increases with more human interactions, it is unclear how effective the proposed methods are since the authors mainly compared with variants of their approach.
> > > > >
> > > > > 2. REINFORCE as a baseline. The authors commented in the rebuttal that it is generally hard to get it working. However, I still think this can be an important baseline, even if it doesn't work. This is especially so because it naturally addresses (or suffers from) the sparse reward problem that the authors were trying to resolve (with the reward propagation).
> > > > >
> > > > > 3. Even simpler reward mapping. In the previous comment, the authors mention mapping real-time feedback is one of the key and novel contributions as part of their whole learning process. Therefore, I believe it's important to analyze this more. Currently, the proposed mapping variants can only assign rewards to 60% (simple) and 80% (propagated) of the actions, which is a also concern for reviewer 4ZhD. A simpler method could be assuming positive unless negative. This assigns reward to all actions, and it seems more natural to the +1 and -1 feedback setting, which is more laborious for the human annotators anyways.
> > > > >
> > > > > 4. Data efficiency analysis. Please correct me if I am wrong, but the SUPONLY baseline seems to match the number of interactions instead of the number of data points. Although the authors show that it's similar to supervised learning,  it's possible that learning with human feedback requires significantly more data samples, and further analysis may be helpful.
> > > > >
> > > > > Overall, I believe that the proposed learning process is not particularly novel, and Figure 4 suggests that the various proposed designs make little difference to learning outcomes and that the key is indeed to get more data. Therefore, I maintain my current evaluation.

---

> > > > > > ### Author Response · Authors · 2023-08-17
> > > > > > **Reply to the reviewer:**
> > > > > >
> > > > > > We are quickly responding here per the points raised by the reviewer. We rushed this response, so we apologize for any typos or grammar errors. We thank the reviewer and the AC for the discussion.
> > > > > >
> > > > > > 1. If we understand the reviewer’s suggestion, it is to collect the same overall number of interactions with the initial system, and do a single round of training. This approach misses on earlier (in the sense of using fewer interactions) opportunities to learn, which means improving earlier, and providing later users a better experience, the way our more rapid improvement cycle does. Technically, this manifests in much higher regret over the lifetime of the system. This is because the executions in this data will be of relatively lower quality, per the performance of the initial system. Therefore, it’s not clear to us how this is an upper bound. The reviewer might mean just using gold-standard demonstrations (as we derive from human-human interactions). We conduct this experiment (SupOnly), showing our much cheaper data (because it requires no annotation or human-human WoZ interactions) is at least as good.
> > > > > >
> > > > > > 2. Our objective becomes identical to REINFORCE if you remove the clipped inverse propensity scoring (IPS) coefficient. However, removing the IPS term with negative rewards immediately reveals the problem with the objective, because you will have terms with negative sign, which means they have unbounded value as we minimize the loss (as probability of examples with negative reward goes to zero, the log probability goes down to $-\infty$). This completely breaks down the optimization. IPS fixes this by weighing down such terms to 0, as their probability decreases.
> > > > > >
> > > > > > 3. We actually found densifying the rewards is not critical. Our heuristics are more complex than suggested, and have little noise (<10%). The proposed approach (assuming positive unless negative) is simpler, but will also generate much more noise. While we didn’t experiment with this approach, our experiments clearly suggest it will be weaker. Can the reviewer explain why this approach is more natural? Studies in human feedback (outside of CS) usually rely on signals in both directions (e.g., Krauss et al. 1966).
> > > > > >
> > > > > > 4. The number of data points in SupOnly is actually higher (we rushed this response, so couldn’t compute exact statistics) than in RewardProp. The reason for that is that human-human games have a much higher score, and the number of instructions correlates well with the score. So human-human games are longer, giving more instructions. We choose to maintain the number of interactions equal to be the most strict towards our approach.

---

> > > > > > > ### Comment · Reviewer_zvAL · 2023-08-18
> > > > > > >
> > > > > > > 1. My original idea was that you would collect the dataset as per the current approach for 11 rounds. After the interactions, further, finetune the model using all collected data. My thought was to compare continual learning and offline learning. However, after taking a closer look at the algorithm, it appears that it is already taking the union of all the previous data, and it essentially re-trains the model using full data in every iteration. I feel this is not exactly true to the continual learning setting, where keeping all previous data is usually not feasible. I would be interested in seeing how the algorithm performs if it only retains, for example, one iteration of the training data.
> > > > > > >
> > > > > > > 2. Even if it does not work, I think showing that will be beneficial, as it further strengthens the author's decision to stick with the contextual bandit setting. An alternative would be PPO-style clipping, which I believe is also suggested by other reviewers.
> > > > > > >
> > > > > > > 3. My suggestion does not contradict the studies on human feedback that the authors cited. In both my suggested setting and the author's setting, there are positive and negative rewards.
> > > > > > >
> > > > > > > It's more about the design of the setting. Asking the annotators to give both positive and negative feedback may cause inconsistencies among them, and this is partially shown as only 60% of the actions have rewards in the simple setting. In fact, another analysis could be consistency analysis between annotators in terms of when they provide feedback. On the other hand, the suggested new setting could alleviate this, making human annotation simpler and more consistent.
> > > > > > >
> > > > > > > Densifying reward being not critical feels somewhat contrary to the paper. For example, the authors proposed a Heuristically Propagated Reward to improve reward density from 60% to 80%.
> > > > > > >
> > > > > > > 4. Thank you for the clarification. I feel this is a surprising result that gold instruction is as effective as binary feedback. It's possible that, as reviewer 4ZhD suggested, it's very much like supervised learning, perhaps because the environment is too simple with only a few possible actions. This somehow invalidates the learning-from-feedback setting, and a more reasonable result would be binary feedback being less effective. I feel a detailed analysis here may be beneficial for the paper.
> > > > > > >
> > > > > > > Again, I'd like to thank the authors for the discussion and their detailed responses.

---

> > > > > > > > ### Author Response · Authors · 2023-08-21
> > > > > > > > **Quick followup -- thank you for the discussion!**
> > > > > > > >
> > > > > > > > Thank you for the discussion. Unfortunately, the lead author is traveling, so we will respond when they return, if the discussion is still open. We just didn't expect the discussion to continue beyond the original Aug 16 deadline (although we appreciate the opportunity).
> > > > > > > >
> > > > > > > > At the meta level, I think there are a few points important to consider. We all seem excited about the prospect of adaptive systems that improve over time from interaction with their users, as we do in this paper. This is extremely understudied (in a transparent way), especially when considering actually building and deploying such systems to interact with humans. Does our paper answer all possible questions in this space? Does it attempt to say the final word? Of course not. This is not possible for almost any interesting ML problem. This is even more challenging when we consider the cost of experiments, both time and money. It's not like doing another run on GPUs that we already have in the lab. We invested > 30k USD in the experiments we show, and > 20k USD in pilot studies that went nowhere. (Yes, we tried more complex algorithms, including variants of PPO, but getting them to work required more careful tuning, which is costly with humans in the loop -- the results are too rudimentary to report with authority, so we can't say anything final). There's also the time issue. Working with MTurk is really hard, and creating a pool of high quality workers to do this relatively complex and time-consuming crowdsourcing task has been an immense challenge.  We don't have an org with internal annotators at our disposal.
> > > > > > > >
> > > > > > > > So, our work shows a working method, which is pretty much a first in this space. Will others identify other approaches that do better? I am pretty certain. But, setting such a fertile stage for future work, well, I think it's a good thing. Are there more experiments we can run? Of course (and much of it forms exciting directions for future work), but there's a cost-benefit calculation that is very different that when just doing yet another run on your GPUs. As a community, we need an avenue to do this type of research that is not closed forever in the labs of billion-dollar companies. Will each such paper answer all possible questions? I don't think that's possible. But, can we learn a lot from such papers? Definitely! I think there's a ton to learn from our paper, and this is what drives whatever it's the community builds in the coming years. I can't wait to see it.

---

> > > > > > > > > ### Comment · Reviewer_zvAL · 2023-08-21
> > > > > > > > >
> > > > > > > > > I also appreciate the authors for the in-depth discussions. Thank you!
> > > > > > > > >
> > > > > > > > > I agree that the authors are working on an interesting problem, and I fully acknowledge that the experiments are difficult and costly. (I also mentioned this as one of the strengths of the paper.) However, I believe this perhaps makes it even more crucial to have rigorous studies, because a potential mistake may lead to a significant amount of money wasted for people that follow this work. This is especially the case for the sample efficiency analysis for example, because it does not require additional human annotation.
> > > > > > > > >
> > > > > > > > > Overall, I think I could have been a bit harsh on the details and would like to raise my score from borderline reject to borderline accept. I hope the lead author can enjoy a well-deserved vacation for completing this work :)

---

### Author Rebuttal · Authors · 2023-08-09

We thank the reviewers for their comments and questions! We look forward to continuing the discussion here during the discussion period. Below are responses to general comments.

Experimenting with different learning paradigms: While we use an objective based on the popular policy gradient REINFORCE objective, we use it in a new learning process that is a new contribution of this work (i.e., learning through live interactions with human users who provide feedback in real time), and demonstrate its efficacy. There are many avenues for future work, including trying more complex RL algorithms such as off-policy learning algorithms or PPO. We opted for simplicity here, focusing on sample efficiency, because of the complexity and costs of studies with human users. And, mostly because this kind of problem is relatively understudied, so we decided to start simple. This is a first step, and we hope our work paves the way for more work in this exciting area, which has so many interesting open problems!

Reproducibility: Conducting these learning experiments at scale (including addressing technical challenges involved in successful deployment and training, but also gaining insights from their deployment, e.g., showing the robustness of our approach to system variations,and analyzing user adaptation over time) and their results are key contributions of our paper. While learning from interactions with human users does pose challenges for reproducibility (i.e., from recruiting and managing a user base), these challenges are unavoidable when studying such complex interaction scenarios. For example, relying on manually designed user simulators will place assumptions on how interactions proceed, require significant engineering effort, and will not reflect the dynamics that arise in real human-agent interaction over time. Indeed, we put significant engineering effort.We are releasing everything, including our results, data, and platform (its original version was released as CerealBar in the past, but we modified it to capture feedback and conduct our studies), all under the permissive MIT license. This is another contribution of our paper: a platform that enables conducting such experiments, and will save much of the engineering effort we put in for future research. Our code and data is also attached to the supplementary material.

Model: The model design is not a main contribution of our work, so we put details in the Appendix. Our model is a modification of the neural architecture used by previous work (Suhr et al. 2019); we chose to adapt an existing model that has been shown to work on this domain, and focus our main contributions on the learning problem itself.

---

### Comment · Reviewer_d6RP · 2023-08-17

I am not able to update my score in my review in openreview system (submit button is always pending somehow), I want to make sure AC and other reviewers know I intend to update my score from 7 to 8. (I will keep trying though)

---

### Decision · Program_Chairs · 2023-09-21

**Decision:**

Accept (spotlight)

**Comment:**

First, the final scores of the paper are 5, 5, 7, 8, 8, with an average of 6.6. Reviewer d6RP intended to update the score to 8 but was unable to do so due to the system.

The work designs a contextual bandit learning approach where human users instruct an agent using natural language.

This is a timely work studying human feedback to improve agent models. Reviewers recognize the following strengths of the work
* Important and timely problem to study.
* Thorough evaluation, with strong empirical results.
* The promise of open-sourcing and data sharing.

We thank the authors and reviewers for the engaging discussions. Most of the concerns of the reviewers were addressed in the rebuttal, and 4 reviewers raised their scores.

For weakness, multiple reviewers pointed out the justification of the 1-step contextual bandit style reward vs. other possible approaches, such as full RL-based methods. The authors opted for simplicity in this work. Although it would be nice to further study the choice of the learning paradigms, the contributions are still meaningful and significant. Please add the further discussions regarding this in the final version.